# Supervised Parametric Learning in the Identification of Composite Biomarker Signatures of Type 1 Diabetes in Integrated Parallel Multi-Omics Datasets

**DOI:** 10.3390/biomedicines12030492

**Published:** 2024-02-22

**Authors:** Jerry Bonnell, Oscar Alcazar, Brandon Watts, Peter Buchwald, Midhat H. Abdulreda, Mitsunori Ogihara

**Affiliations:** 1Frost Institute for Data Science and Computing, University of Miami, Coral Gables, FL 33146, USA; m.ogihara@miami.edu; 2Diabetes Research Institute, Miller School of Medicine, University of Miami, Miami, FL 33136, USA; o.alcazar@med.miami.edu (O.A.); bhw23@miami.edu (B.W.); pbuchwald@med.miami.edu (P.B.); 3Department of Molecular and Cellular Pharmacology, Miller School of Medicine, University of Miami, Miami, FL 33136, USA; 4Department of Surgery, Miller School of Medicine, University of Miami, Miami, FL 33136, USA; 5Department of Microbiology and Immunology, Miller School of Medicine, University of Miami, Miami, FL 33136, USA; 6Department of Ophthalmology, Miller School of Medicine, University of Miami, Miami, FL 33136, USA; 7Department of Computer Science, University of Miami, Coral Gables, FL 33146, USA

**Keywords:** biomarker signatures, early diagnosis, integrated analysis, lipidomics, machine learning (ML), multi-view architecture, multi-omics, metabolomics, parametric models, prevention, proteomics, transcriptomics, type 1 diabetes (T1D)

## Abstract

Background: Type 1 diabetes (T1D) is a devastating autoimmune disease, and its rising prevalence in the United States and around the world presents a critical problem in public health. While some treatment options exist for patients already diagnosed, individuals considered at risk for developing T1D and who are still in the early stages of their disease pathogenesis without symptoms have no options for any preventive intervention. This is because of the uncertainty in determining their risk level and in predicting with high confidence who will progress, or not, to clinical diagnosis. Biomarkers that assess one’s risk with high certainty could address this problem and will inform decisions on early intervention, especially in children where the burden of justifying treatment is high. Single omics approaches (e.g., genomics, proteomics, metabolomics, etc.) have been applied to identify T1D biomarkers based on specific disturbances in association with the disease. However, reliable early biomarkers of T1D have remained elusive to date. To overcome this, we previously showed that parallel multi-omics provides a more comprehensive picture of the disease-associated disturbances and facilitates the identification of candidate T1D biomarkers. Methods: This paper evaluated the use of machine learning (ML) using data augmentation and supervised ML methods for the purpose of improving the identification of salient patterns in the data and the ultimate extraction of novel biomarker candidates in integrated parallel multi-omics datasets from a limited number of samples. We also examined different stages of data integration (early, intermediate, and late) to assess at which stage supervised parametric models can learn under conditions of high dimensionality and variation in feature counts across different omics. In the late integration scheme, we employed a multi-view ensemble comprising individual parametric models trained over single omics to address the computational challenges posed by the high dimensionality and variation in feature counts across the different yet integrated multi-omics datasets. Results: the multi-view ensemble improves the prediction of case vs. control and finds the most success in flagging a larger consistent set of associated features when compared with chance models, which may eventually be used downstream in identifying a novel composite biomarker signature of T1D risk. Conclusions: the current work demonstrates the utility of supervised ML in exploring integrated parallel multi-omics data in the ongoing quest for early T1D biomarkers, reinforcing the hope for identifying novel composite biomarker signatures of T1D risk via ML and ultimately informing early treatment decisions in the face of the escalating global incidence of this debilitating disease.

## 1. Introduction

Type 1 diabetes (T1D) is an autoimmune condition characterized by the destruction of insulin-producing beta cells within the islets of Langerhans in the endocrine pancreas. While the autoimmune nature of T1D is well established, the precise factors triggering this autoimmunity are not fully understood [1,2,3]. Genetic predisposition linked to certain human leukocyte antigen (HLA) genes and other risk factors, such as viral infections and dietary influences, are believed to contribute to the disease onset. Understanding the etiology of T1D is critical for developing preventative therapies and improving the early detection and management of the disease, particularly in children and young adults who represent a significant proportion of the patient population.

Currently accepted biomarkers for T1D include genetic susceptibility markers (e.g., HLA-DRB1 and HLA-DQB1), the presence of autoantibodies against pancreatic islet antigens, and various metabolic indicators (e.g., body mass index, BMI, and glucose tolerance under challenge) [4,5,6]. While they are important clinical indicators of T1D, these biomarkers often have low predictive value, and some manifest late in the disease progression, thereby limiting their promise in preempting the irreversible beta cell damage that heralds the onset of symptoms and clinical diagnosis of T1D [7,8]. Consequently, there is an urgent need for early-detection biomarkers that can accurately discriminate between individuals who will progress to clinical T1D and those who will not. This is particularly crucial given the associated severe long-term complications, which significantly impact life expectancy and quality of life. Despite significant improvements in the management of T1D, patients still lose one to two decades of life-years and have three times higher all-cause mortality compared to the general population [9].

Recent advancements in multi-omics—encompassing proteomics, metabolomics, lipidomics, and transcriptomics, among others—offer promising avenues for identifying new composite biomarker signatures that reflect the complex pathogenic processes of T1D. Previous research has suggested that individual omics approaches provide fragmented insights, which underscores the potential benefits of an integrated multi-omics strategy [10,11]. Such an approach promises a more comprehensive view of the molecular disturbances in individuals at high risk for developing T1D, potentially revealing composite biomarker signatures indicative of disease progression.

Our previous work adopted this integrated multi-omics approach using parallel samples from the same individuals to uncover composite biomarker signatures by analyzing metabolomic, proteomic, lipidomic, and transcriptomic data simultaneously in a cohort of subjects at risk for T1D and healthy controls [12]. These studies highlighted the potential of multi-omics to identify candidates for early biomarkers and elucidate T1D pathogenesis. They also brought to light significant logistical and computational challenges with respect to analyzing the data under such an integrated approach: (1) due to the steep costs associated with longitudinal clinical monitoring as well as data collection across multiple omics, the integrated parallel dataset is inherently small scale in terms of subject count; (2) deciphering potential interactions among measurements within the feature-rich quadra-omics context is combinatorically prohibitive and therefore beyond manual search or filtering. For instance, a dataset containing 100 measurements per 4 different parallel datasets would require examining all possible subsets of features of size *N*, where *N* is the size of the target feature set to be evaluated collectively. This would translate to 400N different feature combinations, which becomes exceedingly large as *N* increases; and (3) the high-dimension nature and low sample size of this dataset imply that a direct application of a machine learning (ML) methodology for feature extraction would be prone to poor performance [13]. Any combination of features identified by a poor-performing model would bear minimal statistical significance and, likely, biologically when examined downstream in the disease context.

Recent works have examined the use of ML methodologies in the context of bioinformatics to combine complementary knowledge brought by different omics while overcoming issues of dimensionality and sample size. A study by Yang and colleagues demonstrated the predictive capacity of multi-omics in colorectal cancer (CRC) survival via ML, which helped to identify key biomarkers for the prediction of CRC patient survival [14]. Such approaches have also been applied in cancer and radiation resistance research and are gaining traction in the literature [15,16,17,18]. These research directions are promising; however, Picard and colleagues noted the challenges of developing meaningful ML models from integrated multi-omics datasets and that a simple concatenation of different omics over a limited biological sample is subject to the “curse of dimensionality” [19]. Such integrated learners have been found to result in worse performance than individual learners in omics data, in addition to increasing the complexity of the problem. Data integration strategies were key to leveraging the potential of multi-omics in this setting, and different stages of integration were proposed. The level of integration is dependent on when ML models are conditioned on features across different omic layers or if results from individual omic learners are integrated post hoc in an ensemble-based scheme. Several proposals, especially in the domain of single-cell analysis, have made predominant use of unsupervised learning methods while evaluating different levels of integration [20,21]. These proposals show a significant leaning toward intermediate integration, where multiple omic layers are analyzed jointly, in addition to incorporating joint dimension reduction and feature selection techniques and robust pre-processing at the individual omic layer. Within the T1D arena, Tan and colleagues proposed the use of supervised ML techniques (e.g., logistic regression, support vector machines, gaussian naïve Bayes, and random forest) to screen signatures from metabolomics, lipidomics, and microbiome data [22]. Their approach was applied to a T1D cohort in Asian populations and targeted healthy and new-onset T1D subjects.

Supervised and interpretable ML approaches have been shown to be especially advantageous in bioinformatic applications where the number of predictors greatly exceeds the number of observations, and understanding how the model arrived at its predictions is also crucial [23,24,25]. To this end, parametric modeling approaches like logistic regression have facilitated the identification of potentially important biomarkers, seen use in diabetes research, and are now established methods in the bioinformatics literature [26,27,28,29].

The current paper proposes an ML framework that can analyze an integrated parallel multi-omics dataset from subjects at a high risk for developing T1D characterized by the high-dimension and low sample-size (HDLSS) regime (see Figure 1) [21]. Adopting a similar integration methodology as that of Picard and colleagues [19], our proposed approach devises statistical models at different stages of integration to evaluate at which stage models can learn under conditions of high dimensionality and variation in feature counts across different omic combinations. These models are applied with the purpose of detecting patterns among high-risk subjects and controls that suggest stratification within this cohort and ultimately extracting salient combinations of biological features across datasets from the different omics that are associated with T1D risk, which can then be subsequently evaluated in future studies for further biological validation. This places our interest squarely in applying statistical models under the supervised learning setting. The framework is analogous to a Knowledge Discovery in Data (KDD) process in the data mining field [30].

More specifically, our approach is designed to answer the following research questions: (1) is it possible to improve classification accuracy with respect to a small cohort, and, if so, at which stage of integration can a supervised learner deliver optimal performance in the parallel quadra-omics dataset; (2) can supervised ML approaches be employed that not only predict with high accuracy but can also reveal salient signatures in the data in association with T1D risk; and (3) do such signatures exhibit enough statistical signal to warrant further scrutiny in the disease context? Via this meticulous ML-based analysis, our long-term goal is to uncover a suite of reliable biomarkers that, after extensive validation, may revolutionize early diagnosis and intervention strategies in T1D, helping to shift the paradigm from damage control to preemptive care.

## 2. Materials and Methods

### 2.1. Sample Collection

Blood samples were obtained at the University of Miami’s Diabetes Research Institute from individuals at increased risk for Type 1 Diabetes (T1D high-risk) as part of TrialNet’s Natural History Study (Pathway to Prevention Study) TN-01 under ancillary study (study ID 195) approved by TrialNet’s IRB, as detailed in our prior work [12]. Roughly 20 mL of blood (in EDTA) was drawn from four high-risk subjects; notably, one subject was identified as a new-onset T1D patient during a second blood sample collection two weeks after the initial sample. For the sake of the machine learning study conducted here, the samples from this individual were excluded. Control samples were also collected from four healthy individuals under a separate IRB-approved study (number 11995-115). All plasma samples were immediately processed and stored at −80 °C. Further demographic and clinical details are also available in [12]. The obtained parallel quadra-omics datasets (proteomics, metabolomics, lipidomics, and transcriptomics) are derived from the same individuals. Any measurements expressed in logarithmic scale were converted to decimal scale before being inputted into our ML framework. Ethical adherence to the Declaration of Helsinki and Good Clinical Practice guidelines was ensured.

### 2.2. Preprocessing Module

In the preprocessing module of the proposed ML framework, the parallel quadra-omics dataset derived from the 9-sample cohort was further refined. To align with the focus on predicting T1D risk before disease onset, we excluded samples identified as new-onset T1D, thereby narrowing the dataset to 7 samples for a targeted analysis contrasting “control” and “high-risk” groups (4 “controls” and 3 “high-risk” subjects).

To construct a complete dataset for the subsequent supervised learning experiments, any features with missing data incidence across the four multi-omics datasets were removed. Given the highly limited sample size with yet a wide abundance of features across the different omics, we posited this approach to be reasonable for the experiments considered here. The curation process resulted in the datasets being reduced as follows: proteomics from 2330 features to 1714 features, metabolomics from 238 to 122, and lipidomics from 66 to 65. The transcriptomics feature count remained unchanged at 329 miRNAs. We found that feature representation from the four omics was comparable among the original and curated datasets. The metabolomics dataset exhibited the highest incidence of missing values, leading to approximately 51% of its features being discarded. The proteomics dataset maintained 74% of its initial features, while lipidomics and transcriptomics datasets were largely unaffected, retaining 98% and 100% of their features, respectively. The final curated integrated quadra-omics dataset encompassed 2230 features representative of the 7 subjects under study.

To gauge the predictive power of supervised learners on the multi-omics T1D high-risk cohort, additional datasets were prepared representing all possible combinations of the individual omic datasets at the 1-omic, 2-omics, 3-omics, and the full 4-omics level. Given the four distinct omic datasets, 15 unique datasets were generated for evaluation in the supervised learning experiments. This approach was designed to investigate the contribution of each omic layer to the classification task and whether a multi-omics approach could reveal significant features that might be overlooked in single-omic analyses and improve prediction accuracy.

For the integration of these datasets into the predictive modeling stage, we adopted three strategies (Figure 1). The method of integration represents a crucial step in managing the high dimensionality of the data and, in turn, improving classification performance. First, an early integration strategy involved a simple concatenation of the multiple omic datasets into a single dataset for supervised modeling, which did not include any further preprocessing of features derived from the different omics. Second, in intermediate integration, the multiple omic datasets were also jointly analyzed with the addition of feature selection techniques, i.e., shrinkage-based and regularization methods, that refined the feature set used for learning while maintaining model interpretability. Third, a late integration strategy was implemented that performed independent analyses at each omic layer and then combined these results post hoc to reach a final consensus. Each integration strategy was evaluated for its ability to enhance classification accuracy and to leverage the comprehensive and interconnected nature of the parallel multi-omics datasets.

### 2.3. Data Modeling

We evaluated the integrated multi-omics datasets generated by the preprocessing module, as presented in Section 2.2, using multiple supervised learning approaches. The current ML framework incorporated parametric modeling approaches capable of retrieving features associated with T1D risk that can later be used to identify candidate biomarkers in a composite signature in future studies [26,27]. This was implemented using logistic regression models, and both its standard and shrinkage-based forms were applied. The former (LR) was used to evaluate classification accuracy obtainable at the early integration stage. The latter was applied likewise but at the intermediate integration stage. For shrinkage-based models, we applied logistic regression with L1 (LASSO) and L2 regularization (RIDGE) to penalize large coefficients such that these were reduced to 0, thereby eliminating redundant and less contributive features. We systematically varied the penalty parameter λ over a coarse grid search. This process incrementally reduced the number of non-zero weight estimates, aiming to retain approximately 20% of the feature space after model training, which concentrated on the most informative predictors for T1D risk.

At the late integration stage, a multi-view model was implemented, following the ensemble scheme shown in the work by Adossa and colleagues [20]. This model integrated individual parametric model learners tailored to each omic layer. The final predictions from the omic-specific models were then aggregated to yield a consensus classification output depending on whether the mean prediction exceeded a set threshold amount *θ*. In this framework, this constant was set to 0.5, or 50%. A diagrammatic overview of its architecture is shown in Figure 2. A rationale of this ensemble approach is that the integration of multiple supervised learners, each trained independently on individual omic datasets, can better leverage diverse biological signals from limited samples, which, consequently, improves prediction accuracy when compared to what any single omic model might achieve on its own. Three different multi-view models were implemented as part of this ML framework. The difference between these models was the choice of classifier used to realize the individual base learners. The first version applied the standard form of logistic regression (MULTI-VIEW LR), which learns a classification model for T1D risk conditioned on the full feature space of an individual omic dataset. The second and third versions used regularized models that applied additional pre-processing with respect to each layer. These versions use logistic regression with L1 penalty (MULTI-VIEW LASSO) and L2 penalty (MULTI-VIEW RIDGE), respectively, and each applies the same grid search strategy as described above to fine-tune the penalty hyperparameter.

Next, we devised a validation methodology using Leave-One-Out Cross-Validation (LOOCV) to evaluate each of the predictive models. Given the 7 T1D high-risk and control subjects from the target dataset, LOOCV was applied to generate 7 independent testing sets, such that each sample was used for testing exactly once while the remaining samples formed the training set. With respect to the training set, each omic feature was standardized to have a mean of zero and a standard deviation of one. The standardization step prevented the possibility of features with larger scales from dominating those with smaller scales. Next, to address the limited training set size (6 subjects) and enhance the robustness of the devised models, we employed smoothed bootstrapping [31]. This method oversamples subjects uniformly at random, where values for each feature across the samples are adjusted by a covariance matrix. The covariance matrix controls the dispersion of newly generated samples in the augmented dataset such that new samples generated do not overlap with existing ones, thereby maintaining the diversity and representativeness of the dataset. The amount of adjustment made using this matrix can be toggled with a shrinkage hyperparameter. The resampling approach taken here is built upon our previous work on developing novel computational approaches for data imputation and amplification from limited parallel multi-omic datasets [32]. In the present study, this technique was used to generate augmented training sets that included 500 control and 500 T1D high-risk virtual subjects/observations derived from the original subjects in the training data.

Logistic regression models and LOOCV were implemented using the sci-kit-learn software package (version 1.3.0) in Python [33]. The smoothed bootstrapping technique was executed using the imblearn package (version 0.0) in Python [34]. All experiments were performed on a 24-core Intel Xeon CPU (2.40 GHz) with 126G RAM.

The multi-view ensemble scheme and experiment codes were also implemented in Python and made publicly available via our project GitHub repository (https://github.com/jerrybonnell/multiomics-t1d-ml (accessed on 31 January 2024)).

## 3. Results

In the present study, we sought to examine the predictive capacity of supervised machine learning models on parallel multi-omics data from a limited T1D high-risk cohort and determine the most effective stage of data integration for learners to identify T1D risk. We also aimed to pinpoint features that are not only salient to the classification but may also serve as candidate biomarkers of T1D risk, subject to subsequent biological validation in future work. Therefore, our evaluation approach is twofold: first, an empirical analysis is performed with respect to the various models’ ability to discriminate between case (high-risk) and control (healthy) subjects, with special attention given to samples that proved challenging to predict; and second, said models are then examined for feature yields, importance, and consistency. In the interest of transparency of the results presented, all reported findings were generated using toolkit suggested hyper-parameters unless noted otherwise.

### 3.1. Empirical Evaluation of Prediction Performance of T1D Risk

Our first approach interrogated the supervised learning classifiers across all combinations of the parallel multi-omic datasets using the LOOCV experiment, as described in Methods Section 2.3. For a specific omics combination, the evaluation made repeated calls of LOOCV to account for the variability introduced by the data augmentation strategy within the training set. In the experiments presented here, LOOCV was repeated 20 times. The presented multi-view models were not subject to combinations where only one omic dataset was present.

We quantified model performance by reporting the mean prediction accuracy and standard deviation of the accuracy scores over the different repetitions. Simultaneously, we also examined observations that were especially challenging for prediction across different combinations of the four parallel omic datasets. In total, among 6 different learning paradigms (LR, LASSO, RIDGE, MULTI-VIEW LR, MULTI-VIEW LASSO, and MULTI-VIEW RIDGE) and 15 omics combinations (4 single omics, 6 dual-omics, 4 triple-omics, and 1 quadra-omics), we trained and evaluated 1560 models. Note that, due to its triviality, multi-view learning paradigms were not subject to datasets consisting of only one omic dataset. Therefore, 240 different models were devised from 3 (non-ensemble) learners at the one-omic level and 1320 models from a total of 6 learners for combinations where at least two or more omic datasets were present. Table 1 presents the results of this experiment.

At the individual omic level, we observed varied performance among the early and intermediate integration learners. Superior performance was exhibited by RIDGE on the lipidomics (L) dataset (86%), which was also the highest obtainable accuracy on this cohort observed across all combinations and omics levels. This same model achieved top performance on all individual omic datasets. At the dual-omics level, other than the proteomics (P) + transcriptomics (T) dataset, late integration models improved the mean prediction accuracy over the early and intermediate models tested. MULTI-VIEW RIDGE improved the mean prediction over RIDGE by 3% on the metabolomics (M) + L dataset and tied with both the RIDGE and LR for best performance on the P+M dataset. MULTI-VIEW LASSO is also tied with its intermediate integration counterpart, LASSO, on the M+T dataset. MULTI-VIEW LR improved the performance over all early and intermediate integration learners by 6% on P+L and by 22% on L+T, when compared to LR, the top performing non-multi view model on this dataset. Strong results by MULTI-VIEW LR suggested that the coefficient penalization by shrinkage-based methods was not helpful on the dual-omics P+L and L+T combinations. Furthermore, when the L+T combination was joined by the proteomics dataset at the tri-omics level (P+L+T), strong performance was also achieved (84%), but it was marginally outperformed by multi-view models with shrinkage-based penalization, i.e., MULTI-VIEW LASSO (+2%) and MULTI-VIEW RIDGE (+2%). Performance by this model deteriorated when encountering the L+T combination joined by the metabolomics dataset (M+L+T, 60%) as well as other tri-omic combinations that included metabolomic features (P+M+L, P+M+T, and M+L+T).

Moreover, other late integration models achieved strong results at the tri-omics level. MULTI-VIEW RIDGE tied with LR and its intermediate model counterpart, RIDGE, for top performance on P+M+L and improved the mean performance by 1% on M+L+T and by 15% on P+L+T when compared to the best-performing early and intermediate integration models tested on these combinations (RIDGE). The exception to this was the combination P+M+T, where late integration models were outperformed by both RIDGE and LR by +2%. Notably, P+M+T contained the P+T combination, which was a dual-omic dataset found to also be difficult for late integration models.

At the quadra-omics level, we found that late integration models were best able to leverage the full combination of the four omic datasets for T1D risk prediction when compared to other approaches. MULTI-VIEW LASSO achieved a top performance of 86%, the highest obtainable mean accuracy observed on this dataset, which is a 15% improvement over the best-performing model (RIDGE) from early and intermediate integration learners. The MULTI-VIEW LR model, which demonstrated strong results at the dual-omics level, was unable to completely overcome its deterioration in performance after encountering the metabolomic dataset at the tri-omics level. Its performance comparatively improved at the quadra-omics level but was outperformed by other late integration models that included robust feature selection at the individual omic layer, as achieved by MULTI-VIEW LASSO and MULTI-VIEW RIDGE. Overall, we found that, for all omic combination levels beyond individual omics, early integration models (LR) tested could deliver optimal performance achievable on a given combination. However, no combination existed where these models could improve the state-of-the-art when compared to late integration approaches. This also held for intermediate integration models (LASSO and RIDGE). We found that, at the dual- and higher omics levels, late integration models were successful in staying competitive or improving the mean prediction accuracy achieved by early and intermediate integration approaches.

### 3.2. Improvements in Instance-Based Prediction Accuracy

The finding that any model was unable to exceed the best performance achieved among the collection of models tested over different combinations of the four omic datasets suggested that there are certain observations within this cohort that may be especially challenging for classifiers to predict well. As shown in the results presented in Section 3.1, there was no such classifier that could deliver mean prediction accuracy beyond 86%, which corresponds to roughly one out of the seven observations that were consistently misclassified under the repeated LOOCV experiment. In this evaluation, we attempted to quantify the difficulty in prediction by examining mean prediction accuracies with respect to each of the seven observations in the cohort. These accuracy measures are obtained using the same LOOCV methodology applied in Section 3.1, where the score for a given instance was derived by computing the mean prediction accuracy for that instance over the 20 trials of LOOCV. These results were stratified by the 6 learning paradigms tested and the 15 different combinations of the 4 parallel multi-omic datasets. This gave an indication as to whether misclassified subjects were consistent across the different models tested and whether specific cases were difficult for a subset of the learners. Figure 3 illustrates the result using a heatmap matrix for each of the learning paradigms.

We found that, across the different omics combinations tested, the collection of classifiers enjoyed the most success in predicting subject #1 (control) and subject #7 (case). Other than the early and intermediate integration models, when subjected to the proteomics (P) dataset, all classifiers obtained almost perfect accuracy when predicting the status of these subjects. Similar success was found for subject #6 (case), and we found, in general, the prediction accuracy to be relatively higher for cases (subjects #s 5–7) when compared to controls (subjects #s 1–4). The exception to this was subject #5 (case). We found that, across several omics combinations and the full quadra-omics dataset, early and intermediate integration models (LR, LASSO, and RIDGE) were unable to correctly classify this subject as T1D high-risk. There were configurations where said subject was correctly predicted by all three, e.g., L, T, L+T, and M+L+T datasets, but this did not hold for most tri-omic and quadra-omic combinations. Ensemble-based schemes, as shown by the three multi-view models tested, were successful in overcoming this difficulty, especially at the quadra-omics level. 

Among controls, we found that subject #3 was especially challenging to predict for the model collection. Among the 6 different learning paradigms and the 15 omics combinations, there were only 4 such configurations where said subject was predicted successfully with perfect accuracy; these were RIDGE when trained on L, M, and M+L datasets and its multi-view counterpart when also trained on M+L. For such configurations, this came at a trade-off as these models encountered difficulty in predicting subjects #4 and #6. There were signs of prediction success in this instance by LR when subjected to M+L, but this also came at the cost of misclassifying the same subjects #4 and #6. For LASSO specifically, we found that this model was further challenged in predicting subjects #2 and #4, especially when encountering datasets at the tri-omic and quadra-omic levels. This was consistent with our results in Section 3.1, which showed this model to exhibit relatively degraded performance over these combinations.

Multi-view models were successful in improving instance-based classification for subjects found to be difficult by early and intermediate integration approaches. MULTI-VIEW LASSO improved prediction over LASSO on these subjects (#s 2, 4, and 5) at the tri-omics and quadra-omics levels. Subject #5, which was found to be difficult for prediction by early and intermediate integration learners at the tri-omics and quadra-omics levels, saw improvements in prediction accuracy by multi-view model counterparts. At the quadra-omics level, these same multi-view models improved instance-based prediction over early and intermediate integration models, with only one subject of the 7 (subject #3) consistently mislabeled by the three different models. With respect to these multi-view learners, there was only one other configuration (P+L+T) where one subject was misclassified out of the total 7.

### 3.3. Feature Importance and Verification Analysis

In the second phase of our evaluation, we focused on isolating features salient to predicting T1D risk at the quadra-omics level. This was accomplished by identifying and scrutinizing “consistent features”, that is, a subset of features repeatedly selected across the seven folds of the LOOCV experiment. We sought to determine if these features appeared with greater frequency than would be expected by chance alone, which would underscore their potential use downstream as candidate biomarkers for T1D risk. “Consistent features” are defined as those that endure the application of coefficient penalization by modeling approaches with shrinkage-based methods, as per our intermediate and late integration models. These methods compress coefficient estimates toward zero, and consistent features are characterized by those with non-zero coefficients post-shrinkage across all seven LOOCV folds. This consistency indicates a robust association with outcome T1D high-risk status beyond the noise inherent in the modeling process over a restricted data sample. 

To test the significance of these consistent features, a permutation test was conducted. This procedure compares the features from the actual LOOCV experiment against those derived from a null distribution. The generation of this distribution was accomplished by randomly permuting the class labels of the subjects and observing which features repeatedly surfaced as important via the same LOOCV modeling process applied to the real data. This randomized trial was repeated 1000 times to provide a distribution of null consistent feature sets against which the consistent features in the real data could be compared. This permutation test was repeated for each of the intermediate and late integration modeling approaches: LASSO, RIDGE, MULTI-VIEW LASSO, and MULTI-VIEW RIDGE. Finally, the identified consistent features from each of the four models were subjected to further validation. We assessed whether these features withstood a series of fold-change thresholds, ranging from minimal (1) to more stringent (1.1, 1.2, 1.3, 2.0, and 3.0) [32]. Such a multi-tiered validation approach ensured that the features we identified were not just statistically significant but also held potential biological significance with respect to T1D risk. Figure 4 and Table 2 present these results.

Among the intermediate and late integration modeling approaches, we observe that multi-view models (MULT-VIEW LASSO and MULTI-VIEW RIDGE) were successful at flagging a larger consistent feature set when compared to their respective intermediate model counterparts (LASSO and RIDGE). MULTI-VIEW LASSO flagged 31 more consistent features than LASSO, and MULTI-VIEW RIDGE flagged 72 more than RIDGE. This was likely an artifact of the shrinkage-based models used within the late integration scheme; these models were conditioned over smaller feature sets (by analyzing individual omics) when compared to intermediate integration models and thus were better able to consistently flag a larger number of repeated features. Of these features, we found that LASSO raised the largest proportion of transcriptomic features (72%) relative to its consistent feature set. Its late integration counterpart, MULTI-VIEW LASSO, traded off the number of consistent transcriptomic features raised for lipidomic (32.1%) and metabolomic (44.6%) features and was also the model that flagged the largest proportion of features from these omic layers. Likewise, this model yielded the greatest share of proteomic features (17.9%).

With respect to the permutation test conducted, we found that the number of consistent features raised by MULTI-VIEW RIDGE was in the tail of the null distribution generated for this learning paradigm. More specifically, we observed that the overlap found by this model was statistically significant at an approximate 90% significance level (*p* = 0.094 by permutation test). This significance was not observed for the other late integration model tested (MULTI-VIEW LASSO) as well as the two intermediate integration models (LASSO and RIDGE).

To further scrutinize these consistent feature sets, we examined whether their constituent features passed a given fold-change threshold(s). We adopted the same threshold amounts, ϴ, as used previously [32]. For lipidomic features, we found a discrepancy between these sets at ϴ = 1.3 and ϴ = 2.0 before all such features were eliminated completely at ϴ = 3.0. At the former, all RIDGE features were retained, while the set flagged by MULTI-VIEW RIDGE experienced the largest reduction while still preserving 94.3% of its features. At the latter, RIDGE also retained the largest proportion, 42.9%, while all features raised by LASSO were filtered out. Other models retained approximately 20%. For metabolomic features, we found that all such features identified by LASSO were also eliminated completely at θ = 2.0, although the feature set size flagged by this model is relatively small (3 metabolites). Other approaches retained some proportion even at the highest threshold tested (highest 19.4% by MULTI-VIEW RIDGE). A late integration model (MULTI-VIEW RIDGE) yielded the largest proportion of these features at θ = 1.2 and θ = 1.3, which was then outperformed by RIDGE at θ = 2.0 with 43% features retained. With respect to proteomic features, no discrepancy was found between these models until encountering ϴ = 2.0. Proteomic features raised by LASSO survived the remaining thresholds tested; however, as with its metabolomic feature set, the number of consistent proteomic features available is comparatively small (1 proteomic feature). For other models, MULTI-VIEW LASSO retained the largest proportion at this threshold (60%), while MULTI-VIEW RIDGE had the largest reduction and retained 46.2% of the features in its respective set. A large proportion of the features identified by RIDGE did not survive the most stringent threshold tested, with only 28.6% of its features remaining. Other than LASSO, the late integration model MULTI-VIEW LASSO retained the largest proportion of proteomic features at this threshold (30%), while MULTI-VIEW RIDGE was the lowest (23.1%).

Transcriptomic features performed the best with respect to this validation step. The four modeling approaches tested retained their respective features from this omic layer at θ = 1.1, 1.2, and 1.3. At θ = 2.0, the worst performing feature set (from MULTI-VIEW RIDGE, a late integration model) retained 78% of its features and 59% at θ = 3.0. In contrast, the entire consistent feature set identified by MULTI-VIEW LASSO was recovered after being subjected to each threshold amount. Most of the features raised by LASSO also survived the most stringent threshold tested (95% of features retained at θ = 3.0).

## 4. Discussion

The above results indicate that a prerequisite to identifying biomarkers associated with T1D risk via the application of ML is to first evaluate combinations of data integration strategies and learning paradigms that can be used to devise a classifier capable of discriminating between case (T1D high-risk) and control subjects. Such a classifier can then be used to retrieve features from different omic layers that are salient to the classification, which when taken collectively, can form a more comprehensive picture to potentially assess T1D risk. To this end, we developed an ML framework that trains different parametric modeling approaches (LR, LASSO, and RIDGE) on a dataset iteratively subjected to one of three stages of data integration (early, intermediate, and late). This framework is depicted in Figure 1. A fundamental challenge to the approach taken here is the limited sample size of the source parallel multi-omics datasets, which is reflective of the steep cost and logistical challenges of collecting longitudinal biological sample cohorts and performing the associated multi-omics analyses. These costs are also significant from a computational perspective, in which we highlight two key limitations. First, the restricted sample size implies that the currently devised ML models may have limited generalization capabilities, especially when used for validation in independent T1D cohorts. Second, the ML framework must be trained on the high-risk labels that result from data collection. Given that these are derived using current classification criteria for T1D risk, which cannot predict with certainty who will or will not eventually progress to disease onset, the devised models may suffer from propagating current diagnostic flaws. Notwithstanding this second limitation, the proposed ML models are designed to overcome traditional (single-omic) approaches under existing risk-classification criteria to ultimately identify potential novel biomarkers by incorporating multi-omic profiles, which might be overlooked otherwise.

Our results on the comparative performance of modeling approaches under this framework across different combinations of omics data, ranging from single-omic to quadra-omics, showcased the potential utility of a computational approach in this limited setting. Our findings indicate that the highest mean prediction accuracy achieved at the quadra-omics level (86%) is also the highest prediction rate achieved across all omic levels and combinations (Table 1 and Figure 3). This result was also matched by combinations at lower omic levels, e.g., the lipidomics dataset (L) and the combination of the lipidomics and transcriptomics datasets (L+T). While achieving top performance with fewer omic combinations may seem advantageous computationally, the equal performance at the quadra-omics level represents a significant step forward in this research and aligns with the biological objective of creating a holistic view of disease pathogenesis and the identification of associated composite biomarker signatures. The alignment of these two goals is particularly notable given the computational challenges posed by the limited sample size and relatively vast number of features from different omic profiles.

Stratifying prediction scores by observation reveals added conclusions regarding the strengths of individual classifiers tested in the model collection. Analysis of the heatmap matrices, as per Section 3.2, shows that specific omic combinations from certain models are associated with improved predictive accuracy for some subjects. For instance, subject #3 (control) was found to be particularly challenging but was classified perfectly by RIDGE on the metabolomic (M) and lipidomic (L) datasets, as well as their combination (M+L). This finding highlights the potential in future work to leverage certain omic combinations within an ensemble to bolster predictions and improve model training at the quadra-omics level. Such an approach could more effectively capitalize on the use of an ensemble-based learning scheme, as proposed in this study. Depending on whether such fine-tuned models exhibit notable differences in feature importance when compared to the overall ensemble, this can also inform clinically whether there are substantial differences in these instances when compared to other subjects in the cohort.

In our quest to spotlight significant features for downstream biological scrutiny in future work, we observed that varying combinations of features, learning, and integration paradigms contribute to diverse representations from different omic layers in the consistent feature set. To this end, we recognize contributions made by intermediate modeling approaches. Specifically, RIDGE models were successful in flagging a large proportion of lipidomic features from its consistent feature set that passed the given fold change thresholds tested. This also holds for the proteomic feature set derived by LASSO; however, only a small number of such features were available for analysis. RIDGE models were also successful in highlighting strong metabolomic as well as proteomic features at the most stringent threshold values.

The fact that late integration models were the strongest performing, with respect to mean prediction accuracy and consistent feature set size, points to promising avenues for expansion in future work and methods for overcoming limitations of very low sample sizes in multi-omic cohorts. Given that the proposed late integration models are composed of independent learners trained at the individual omic layer, the independent and distributed nature of the learning procedure under this approach implies that the model can scale as more data become available in a specific omic, potentially leading to more accurate and generalizable predictions. This would facilitate a more rapid expansion of the current dataset in future work by relaxing the current requirement on any new samples introduced into the ML framework to improve learning: variables across all four parallel multi-omic datasets must be collected simultaneously. The addition of a larger dataset from a particular omic profile would also lay the groundwork for implementing transfer learning strategies. Under this approach, knowledge gained from one omic domain (e.g., a larger proteomics dataset) can be evaluated for its transfer to another (i.e., the current quadra-omics dataset). This becomes especially critical in overcoming the challenges associated with the high dimensionality and very low sample size in the current cohort, as well as improving generalization capabilities and model robustness when validating the approach in other independent cohorts. By pre-training the model on the larger dataset and fine-tuning it on the smaller, more specific quadra-omics dataset, the model can leverage broader patterns learned from the larger dataset while adapting to the specificities of the integrated parallel quadra-omics data. This would also open opportunities for continuous learning, where such a model can be periodically updated as new data become available, thus keeping the model relevant and improving its performance over time.

## 5. Conclusions

In summary, the integration of diverse omic layers—in this case, proteomics, metabolomics, lipidomics, and transcriptomics—using ML has shown considerable promise in enhancing the accuracy of classifying subjects at high risk for T1D and paving the path toward identifying biomarkers salient to T1D risk in future studies. Our empirical evaluation suggests that ML classifiers can deliver high prediction accuracy in limited cohorts and that performance is dependent on the stage of data integration and whether learners are conditioned on features derived from single or multiple omic profiles simultaneously. This underscores the potential of multi-omics approaches in uncovering complex biological relationships and patterns that might be invisible when analyzing each omic layer in isolation. To this end, we find that late integration approaches, specifically using the multi-view architecture proposed here, are most successful in achieving high prediction accuracy at higher omic levels. A late integration model consisting of individual logistic regression learners with L2 penalty was also found to exhibit a significantly larger consistent feature set when compared to chance models. The fact that these sets of features identified are also able to withstand verification at set fold-change thresholds will be a significant step downstream in recognizing robust biomarkers for T1D risk in future work.

The present study not only demonstrates the feasibility and efficacy of using ML for analyzing integrated parallel multi-omics data in T1D research but also lays the foundation for expanding the ML framework in future work. Future directions include incorporating larger datasets from a specific omic profile as part of a transfer learning scheme, continuous learning, and validating the devised multi-view models in independent T1D cohorts. We are also interested in evaluating the impact of potential biomarkers identified by the proposed ML framework in other populations [35]. Furthermore, quadra-omics is not the upper bound in multi-omics analysis, and the incorporation of additional omic profiles, such as peptidomics and genomics, may prove fruitful in future research. The results of these interdisciplinary endeavors highlight the potential of ML methods in identifying composite biomarker signatures for early detection of T1D and guiding timely intervention for effective prevention, marking a significant stride toward enhanced healthcare outcomes. This has the potential to revolutionize the understanding of T1D pathogenesis and aid the development of more targeted and effective treatment strategies.

## Figures and Tables

**Figure 1 biomedicines-12-00492-f001:**
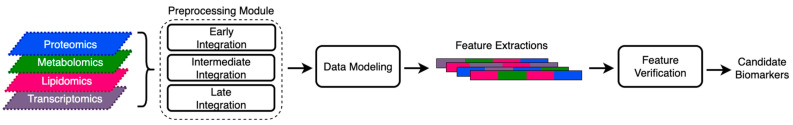
Schematic overview of multi-omics data processing and analysis pipeline. This diagram illustrates the sequential stages of the present multi-omics analysis framework, beginning with the parallel collection of high-dimensional datasets from the four distinct omic layers, namely proteomics, metabolomics, lipidomics, and transcriptomics. These datasets are then directed via a preprocessing module that supports three integration strategies: early integration, where raw data from all omics are combined before supervised analysis; intermediate integration, which also involves jointly analyzing multiple omics datasets but with the addition of variable selection approaches (i.e., shrinkage methods and regularization); and late integration, where independent omic analyses are performed, and their results are combined at a later stage. Following integration, data modeling is employed for the prediction of T1D risk, as well as to discern patterns and interactions within the integrated dataset, leading to the extraction of pertinent features. The subsequent feature extraction phase isolates significant variables as candidate biomarkers for later biological validation in future studies. These features are then rigorously verified to ensure their consistency and predictive validity across multiple analytical iterations. The final phase involves the selection of validated biomarkers as part of a composite signature.

**Figure 2 biomedicines-12-00492-f002:**
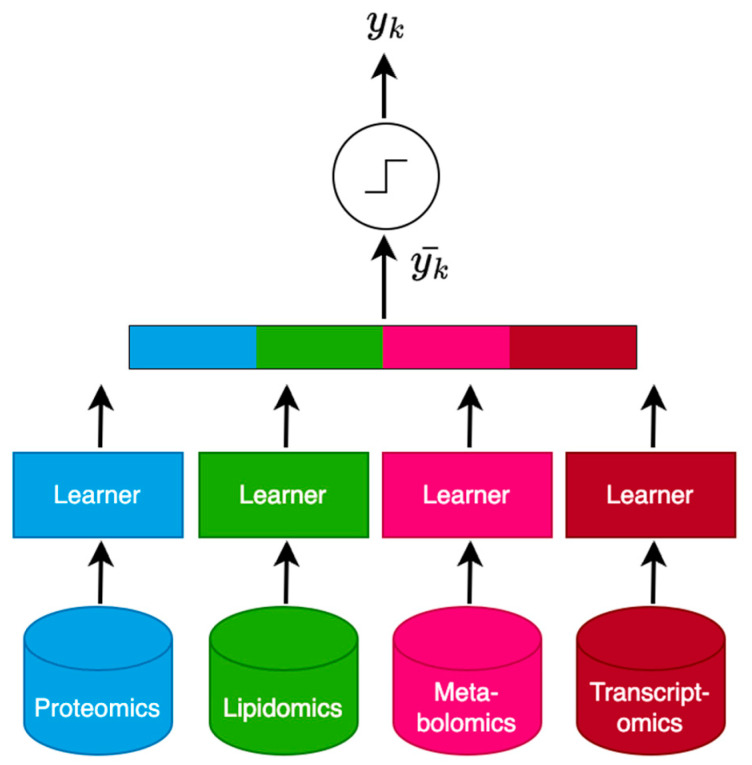
A diagrammatic representation of the implemented multi-view ensemble architecture for learning from the parallel multi-omics T1D high-risk cohort in the late integration stage. Independent machine learning models termed “learners” are trained on the individual omic datasets: proteomics, lipidomics, metabolomics, and transcriptomics. Each learner analyzes their dataset independently to predict an outcome for some sample with index *k*, and their predictions $\hat{y_{k}}$ are aggregated—via simple arithmetic mean—into a consensus prediction for each sample $\bar{y_{k}}$. This ensemble prediction is then subject to a set threshold, *θ*, and predictions surpassing this are classified as “T1D high-risk” or otherwise as “control”. This yields an outcome prediction, *y_k_*. The approach takes advantage of biological insights that can be extracted from each omic layer when analyzed independently and aims to enhance these by aggregating the predictions collectively to scale individual learners to the quadra-omics setting and improve the robustness of the overall model.

**Figure 3 biomedicines-12-00492-f003:**
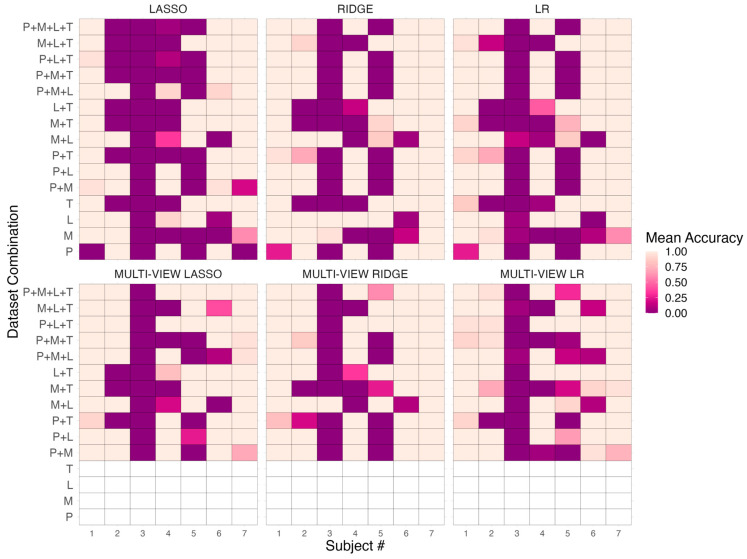
Heatmaps illustrating the efficacy of various predictive models—LASSO, RIDGE, and LR, and their multi-view ensemble equivalents—when applied to different combinations of the parallel individual omic datasets, namely proteomics (P), metabolomics (M), lipidomics (L), and transcriptomics (T). Intensity of coloration corresponds to mean prediction accuracy for a given subject over repeated trials of LOOCV, with gradations of tan and purple colors representing higher and lower prediction accuracy, respectively. Subjects 1–4 were controls and 5–7 high-risk for T1D. Cells in white indicate no prediction for the given model.

**Figure 4 biomedicines-12-00492-f004:**
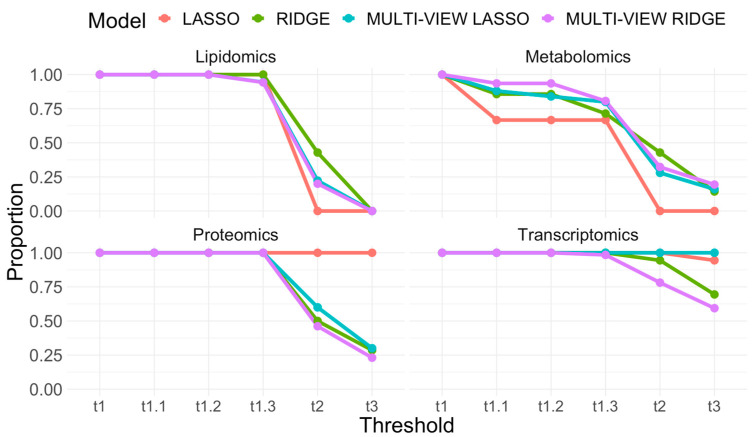
Line plots illustrating the proportion of selected features from the consistent feature sets that surpass given fold-change thresholds for each omic type (1, 1.1, 1.2, 1.3, 2.0, and 3.0). Lines demonstrate how the proportion of features that meet or exceed these threshold values changes as the stringency of the threshold increases. One line is shown for each of the intermediate and late integration models tested—LASSO, RIDGE, MULTI-VIEW LASSO, and MULTI-VIEW RIDGE.

**Table 1 biomedicines-12-00492-t001:** Comparison of the predictive performance of six machine learning models (LASSO, RIDGE, LR, MULTI-VIEW LASSO, MULTI-VIEW RIDGE, and MULTI-VIEW LR) with respect to mean prediction accuracy and standard deviations over repeated trials of LOOCV. Different combinations of the four parallel omics datasets are tested: proteomics (P), metabolomics (M), lipidomics (L), and transcriptomics (T). Each row represents a unique omic dataset or a combination of two or more datasets, and the corresponding values indicate the mean performance and standard deviation of each model for that specific combination. Blank cells denote settings where a model was not tested. Values for best-performing models for a given combination are bolded.

	Repeated LOOCV Prediction Performance *					
Omic	LASSO	RIDGE	LR	MULTI-VIEW LASSO	MULTI-VIEW RIDGE	MULTI-VIEW LR
P	0.43 ± 0	**0.61 ± 0.063**	**0.61 ± 0.063**			
M	0.37 ± 0.072	**0.59 ± 0.044**	0.39 ± 0.094			
L	0.71 ± 0.056	**0.86 ± 0.032**	0.72 ± 0.032			
T	**0.57 ± 0**	**0.57 ± 0**	0.56 ± 0.064			
P+M	0.59 ± 0.064	**0.71 ± 0**	**0.71 ± 0**	0.67 ± 0.067	**0.71 ± 0**	0.54 ± 0.079
P+L	0.71 ± 0	0.71 ± 0	0.71 ± 0	0.75 ± 0.063	0.71 ± 0	**0.81 ± 0.07**
P+T	0.43 ± 0	**0.66 ± 0.07**	**0.66 ± 0.072**	0.56 ± 0.044	0.57 ± 0.093	0.56 ± 0.044
M+L	0.62 ± 0.07	0.7 ± 0.064	0.58 ± 0.086	0.6 ± 0.059	**0.73 ± 0.044**	0.71 ± 0.066
M+T	**0.57 ± 0**	0.56 ± 0.044	0.52 ± 0.096	**0.57 ± 0**	0.46 ± 0.063	0.54 ± 0.112
L+T	0.57 ± 0	0.59 ± 0.052	0.64 ± 0.073	0.69 ± 0.059	0.76 ± 0.07	**0.86 ± 0**
P+M+L	0.69 ± 0.059	**0.71 ± 0**	**0.71 ± 0**	0.58 ± 0.056	**0.71 ± 0**	0.61 ± 0.067
P+M+T	0.43 ± 0	**0.71 ± 0**	**0.71 ± 0**	0.56 ± 0.032	0.69 ± 0.052	0.56 ± 0.064
P+L+T	0.44 ± 0.056	0.71 ± 0	0.71 ± 0	**0.86 ± 0**	**0.86 ± 0**	0.84 ± 0.044
M+L+T	0.57 ± 0	0.7 ± 0.044	0.59 ± 0.064	0.63 ± 0.072	**0.71 ± 0**	0.6 ± 0.075
P+M+L+T	0.44 ± 0.032	0.71 ± 0	0.71 ± 0	**0.86 ± 0**	0.8 ± 0.072	0.75 ± 0.063

* Mean LOOCV prediction accuracy percentages and standard deviations over the 20 trials shown. Values rounded to 3 digits of precision. Blank means no prediction for a given model. Bold indicates best performance.

**Table 2 biomedicines-12-00492-t002:** Summary of different criteria for consistent features identified by the intermediate and late integration models. Criteria listed: number of features repeatedly selected over 7 folds of the LOOCV experiment, *p*-value indicating the statistical significance of the consistent feature set with respect to the permutation test, and percentage contribution of each omic layer to the set—proteomics (%P), lipidomics (%L), metabolomics (%M), and transcriptomics (%T). *p*-values < 0.1 indicate statistical significance.

Model	# Features	*p*-Value	%P	%L	%M	%T
LASSO	25	0.462	4.0	12.0	12.0	72.0
RIDGE	71	0.32	19.7	19.7	9.86	50.7
MULTI-VIEW LASSO	56	0.295	17.9	32.1	44.6	5.36
MULTI-VIEW RIDGE	143	0.094	9.09	24.5	21.7	44.8

## Data Availability

The current study was conducted on parallel multi-omics datasets previously deposited in the following publicly accessible repositories: the NIH Common Fund’s National Metabolomics Data Repository (NMDR; www.metabolomicsworkbench.org, accessed on 31 January 2024), accession #s ST001690 (doi:10.21228/M8B123) for the metabolomics dataset and ST001642 (doi:10.21228/M8ZX18) for the lipidomics dataset; the PRIDE database of ProteomeXchange (https://www.ebi.ac.uk/pride/, accessed on 31 January 2024), accession # PXD023541 for the proteomics dataset; and the Harvard Dataverse repository (doi.org/10.7910/DVN/A2OU24) for the transcriptomics dataset. Code for the multi-view ensemble scheme and experiments conducted in this research can be freely accessed and downloaded in Python via the project GitHub repository (https://github.com/jerrybonnell/multiomics-t1d-ml, accessed on 31 January 2024).

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
