# Peer review of "Supervised Parametric Learning in the Identification of Composite Biomarker Signatures of Type 1 Diabetes in Integrated Parallel Multi-Omics Datasets"

_biomedicines, 2024, doi:10.3390/biomedicines12030492_

Round 1

Reviewer 1 Report

Comments and Suggestions for Authors

The paper is interesting and well written. The authors investigated the risk for developing type 1 diabetes applying machine learning for the purpose of improving identification of salient patterns and novel biomarkers. The results confimed the efficacy of machine learning in exploring integrated parallel multi-omics data in the ongoing quest for early type 1 diabetes biomarkers. One question arises: can the methodology permit to investigate the impact of these biomarkers in elderly patients and if  biomarkers increase the risk of dementia in diabetic patients?  (see and add as references papers by Murdaca et al concerning machine learning in geriatric patients)

Comments on the Quality of English Language

Minor english editing

Reviewer 2 Report

Comments and Suggestions for Authors

I have reviewed your paper. Your paper addresses a timely and important topic and makes a significant contribution to the field of diabetes biomarkers. You have conducted a comprehensive and rigorous analysis of parallel multi-omics data from subjects at high risk for type 1 diabetes (T1D) and applied machine learning models to identify composite biomarker signatures of T1D risk. Your paper is well-written, clear, and concise.

I only have some minor comments and suggestions for improvement, which are listed below:

- On page 7, line 15, you concluded that parallel multi-omics can be used to identify integrated biomarker signatures of T1D risk, and that machine learning can help explore the integrated dataset and extract salient features that may serve as candidate biomarkers of T1D risk. However, you did not acknowledge the limitations of your study, such as the small sample size, the lack of validation in independent cohorts, and the possibility of confounding factors or sources of bias. Please address these limitations in the paper and suggest directions for future research.

Sincerely,

Reviewer 3 Report

Comments and Suggestions for Authors

In this paper, the authors primarily focused on addressing the challenge of identifying early biomarkers for Type 1 diabetes (T1D) using parallel multi-omics approaches and machine learning (ML) techniques. The study aims to improve the identification of salient patterns in integrated multi-omics datasets, ultimately leading to the extraction of novel biomarker candidates. To achieve this, the article evaluated the use of ML methods, incorporating data augmentation and supervised learning, to enhance pattern recognition in the data. However, there are several concerns regarding the study. Firstly, it is unclear whether the multi-omics data are from the same patients. Paired samples are crucial for this type of data integration. If the different molecular omics are from different patients, the proposed methods and results may be limited. Secondly, the sample size used in the study was limited, which may have restricted the model's training and generalization capabilities. Furthermore, the study did not explicitly address the issue of data imbalance, which could potentially bias the model's predictions towards the majority class. Thirdly, the study did not provide detailed insights into feature selection and model interpretability, making it difficult to understand how the model arrived at its predictions. Additionally, the choice of early, middle, or late integration might correspond to the experimental design, not just to the prediction performance. Lastly, although the study identified potential biomarker candidates, these need to be further validated in clinical settings to confirm their effectiveness. This validation process may require additional time and resources, including complex clinical trials and long-term patient monitoring. For referring a closely related reference PMID: 32339126, the proposed method therein of aligning the multi-omics data might bring new insights and improvements to this paper. Although the study identified potential biomarker candidates, further research and validation are needed to confirm its effectiveness and applicability in real-world settings.

Comments on the Quality of English Language

The language need be polished.

Reviewer 4 Report

Comments and Suggestions for Authors

The paper highlights the effectiveness of machine learning in examining integrated parallel multi-omics data to pinpoint early biomarkers associated with Type 1 Diabetes, thereby improving the chances of uncovering new composite biomarker signatures for T1D risk assessment. This offers potential for guiding early treatment decisions in light of the increasing global incidence of this debilitating disease. While the paper's research motivation is evident, it encounters several issues:
1. Although the paper discusses the utilization of ML methodologies in bioinformatics, the specific ML aspect employed by the authors remains unclear. This lack of clarity is evident in the title and abstract.
2. The related work section should be a standalone segment, with a particular focus on reviewing machine learning methods in medical diagnosis, thus emphasizing the research motivation.
3. With only four sections, the paper's overall structure requires optimization, particularly in the experimental section, where a comparative analysis with similar medical data analysis methods is necessary to underscore the method's advantages.
4. The absence of an overarching architecture diagram hinders the illustration of the end-to-end approach.
5. The insufficient number of references compromises the quality of the literature review, especially in identifying specific machine learning methods.
